

**Postmidnight equatorial plasma irregularities on June solstice during low solar activity**
**– a case study**
Claudia M. N. Candido[1,2], Jiankui Shi[1], Inez S.Batista[2], Fabio Becker-Guedes[2], Emília
Correia[2,6], Mangalathayil A. Abdu[2,4], Jonathan Makela[3], Nanan Balan[7], Narayan
Chapagain[5], Chi Wang[1], Zhengkuan Liu[1]
[1]National Space Science Center, NSSC, Chinese Academy of Sciences, State Key
Laboratory, China-Brazil Joint Laboratory for Space Weather, China
[2]National Institute for Space Research – INPE, - São José dos Campos, SP, Brazil
[3]Department of Electrical and Computer Engineering, University of Illinois at Urbana-
Champaign, Urbana, Illinois 61801, U.S.A
[4]Instituto Tecnológico de Aeronáutica – ITA – São Jose dos Campos, Brazil
[5]Departament of Physics, Patan Multiple Campus, Tribhuvan University, Latitpur, Nepal.
[6]Centro de Radio Astronomia e Astrofísica Mackenzie, CRAAM, University Presbiteriana
Mackenzie – São Paulo – Brazil
[7]Institute of Geology and Geophysics, Chinese Academy of Sciences, Beijing 100029,
China

Corresponding author: claudia.candido@inpe.br
*Keywords*: Solar minimum, Spread-F, Postmidnight plasma irregularities, equatorial
ionosphere, ionosonde



**Abstract**
We present a case study of unusual spread-F structures observed by ionosondes at two
equatorial and low latitude Brazilian stations - Sao Luis (SL: 44.2° W, 2.33° S, dip angle:
−6.9°) and Fortaleza (FZ: 38.45°W, 3.9° S, dip angle: −16°). The irregularity structures
observed from midnight to post-midnight hours of moderate solar activity (F10.7 < 97) have
characteristics different from typical post-sunset equatorial spread-F. The spread-F traces
first appeared at or above the F-layer peak and gradually became well-formed mixed spread-
F. They also appeared as plasma depletions in the 630.0 nm airglow emissions made by a
wide-angle imager located at nearby low latitude station Cajazeiras (CZ: 38.56° W, 6.87° S,
dip angle: -21.4°). The irregularities appeared first over FZ and later over SL, giving
evidence of an unusual westward propagation or a horizontal plasma advection. The drift
mode operation available in one of the ionosondes (a Digital Portable Sounder, DPS-4) has
enabled us to analyze the horizontal drift velocities and directions of the irregularity
movement. We also analyzed the neutral wind velocity measured by a Fabry-Perot
interferometer (FPI) installed at CZ and discussed its possible role on the development of
the irregularities.

1  **Introduction**

Equatorial spread-F representing small scale to large scale plasma irregularities has been
extensively studied for several decades. The large-scale plasma irregularities specifically
known as equatorial plasma bubbles (EPBs) are known to be associated with equatorial
spread-F. In the Brazilian equatorial sector, characterized by large negative magnetic
declination, spread-F and EPBs have high occurrence rates during local summer and
equinoctial months (Abdu et al., 1981; Sahai et al., 2000; Sobral et al., 2002). However,





during low solar activity conditions, there is a class of spread-F/plasma irregularities
regularly observed in distinct longitudinal sectors. They are known as post-midnight plasma
irregularities (PMIs), which occur mostly in June solstice. A recent review of plasma
irregularities is provided by Balan et al. (2018).
PMIs occur under conditions considered not favorable for the development of the Rayleigh-
Taylor (RT) instability, since that at night the vertical plasma drifts are downward, owing to
the westward electric fields. In recent years, a variety of works have reported their
occurrence both at low latitudes and equatorial region. Otsuka et al. (2009) and Nishioka et
al. (2012) investigated PMIs over Indonesia and discussed their possible sources. Li et al.
(2011) reported these irregularities observed over Hainan, China during low solar activity.
Candido et al. (2011) presented a study of PMIs observed over the south crest of the
equatorial ionization anomaly (EIA) during low solar activity, in CP, Brazil. Yokohama et
al. (2011) studied unusual patterns of echoes from coherent scatter radar data occurring
around midnight during the solar minimum period. They observed two principal types of
irregularities: the upwelling plumes and MSTID-like striations. They have argued that the
former can be generated by both the RT instability (at equatorial region) or to Perkins
instability (at mid-latitude region) and the later only by the Perkins instability.  Yizengaw et
al. (2013) presented the study of the PMIs over equatorial Africa, and also investigated their
most probable causes. Dao et al. (2017) reported in a very interesting work the occurrence of
postmidnight field-aligned irregularities (FAIs) in Indonesia during low solar activity in

78  2010.

Many instrumental techniques are currently providing high-quality measurements and
results for ionospheric studies. Early investigations of the ionosphere referred to the diffuse
echoes seen in data from measurements using ionosondes, which are high-frequency radars
used for ionospheric sounding (Breit and Tuve, 1926; Booker and Wells, 1938). The

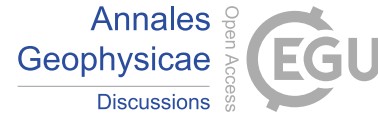

"spread-F" is widely used to generically refer to the irregularities observed in the equatorial
and low-latitude regions. Nowadays, digital ionosondes are extensively used for ground-
based sounding of the ionosphere, providing information from the E-region to the peak of
the F-layer, over a variable range of frequencies as well as features related to the
propagation of the irregularities (Reinisch et al., 2004; Batista et al., 2008, Abdu et al.,
2009). Equatorial spread-F has been extensively studied for several decades, and it is known
to be associated with the occurrence of large-scale plasma irregularities or equatorial plasma
bubbles (EPBs).
Optical imaging of thermospheric emissions, like that used in this work, is also a useful
ground-based technique for studying thermosphere/ionosphere processes. All-sky imaging
systems provide images of thermospheric emissions (e.g., OI 630-nm, OI 777.4-nm
emissions) at ionospheric heights over a large horizontal extent. The OI 630-nm emission
comes from recombination processes between molecular oxygen and electrons and presents
a volumetric emission rate which peaks at an altitude of ~250 km, around the F-layer the
bottom side height. In this way, variations in the intensity of the emission (dark and bright
regions) are used as tracers of ionospheric irregularities, such as EPBs, or other
disturbances, such as travelling ionospheric disturbances - TIDs (Pimenta et al., 2008;
Abalde et al., 2009; Makela et al., 2010; Candido et al., 2011; Chapagain et al., 2012).
For clarity for the present study, which presents a distinct pattern of spread-F from those
usually observed in equatorial ionograms, we first address the current state of understanding
regarding spread-F signatures in ionosonde data.
It is currently accepted that there are two main spread-F types: range and frequency type
spread-F traces (Abdu et al., 1998). The range type spread-F, often associated with the
occurrence of medium and large-scale irregularities, including EPBs, is comprised of trace
patterns with the echoes spread in range and with the onset beginning at the lower frequency



end of the F-layer trace in ionograms. During the spread-F season in Brazil, between
October and March, the evening pre-reversal enhancement in the zonal electric field, and
therefore the F-layer vertical drift, attains large values and range type spread-F is observed
in equatorial ionograms, followed by their appearance at crest region of the EIA, which is
located around Cachoeira Paulista (CP: 22.4° S, 45° W, dip angle: -37°).   During the
remaining part of the year, when the vertical drifts are very small (Batista et al., 1996),
spread-F is restricted to the height region below the F-layer peak, rarely reaching the topside
ionosphere, and therefore observed only close to the dip equator. This type of spread-F is
usually classified as bottom side spread-F (Valadares et al., 1983). The other common
spread-F pattern observed in equatorial ionograms is the frequency type spread-F. In this
case, the spread-F echoes are seen at frequencies around the F-layer critical frequency
(foF2). It is believed to be associated with smaller scale/decaying irregularities following
spread-F/EPBs (Abdu et al., 1981a).
Some studies have pointed out that frequency type spread-F can sometimes be associated
with patches of ionization propagating eastward (MacDougall et al., 1998) and this type is
frequently observed in solstices in distinct longitudinal sectors. Additionally, both frequency
and range spread-F types can appear simultaneously, as a mixed spread-F pattern.
In this work, we present a case study on an unusual/anomalous spread-F/plasma
irregularities/depletions pattern observed over the equatorial region. We use the term
"unusual" in the sense that the observed features are distinct from those typically observed
for spread-F associated with post-sunset spread-F, as described above. Although the unusual
type of spread-F has been recognized since the early studies of the equatorial ionosphere
(Munro and Heisler, 1956; Heisler, 1958; Calvert and Cohen, 1961; Bowman, 2001), this is
the first time that it is reported for the Brazilian equatorial region with simultaneous airglow
observations, which reveal important ionospheric characteristics not available when using





only ionosonde data. The earlier studies extensively reported the occurrence of anomalies in
F-layer traces, such as cusps, F2 forking, and their possible association with TIDs. Calvert
and Cohen (1961) presented a comprehensive study of the distinct spread-F patterns. They
concluded that the distinct configurations or shapes of spread-F were associated with the
scattering in the vertical, east-west plane from field-aligned irregularities and that the
spread-F pattern depends on the position relative to the ionosonde and the scale sizes of the
irregularities.

**2. Data and Method**

**2.1  Digisondes**

We analyzed ionograms from two Digisondes DPS-4 operated at two Brazilian equatorial
sites: SL (44.2° W, 2.33° S, dip angle: −6.9°) and FZ (38.45° W, 3.9° S, dip angle: −16°),
which are separated in the east-west direction by ~600 km. Both instruments provided
ionograms at a 10-minute cadence. The DPS-4 also performs echo directional studies based
on Doppler interferometry, which provides information about the drift velocities associated
with irregularities. The operation of each Digisonde is based on the transmission of pulses at
digital frequencies from 1 to 20 MHz that are reflected from the ionosphere at plasma
frequencies lower than foF2. The maximum height range of the ionograms can be set at
~700 or ~1400 km, for which the resolution is ~5 km and ~10 km, respectively. The
ionospheric true heights are calculated by an inversion method implemented by the ARTIST
software (Reinisch et al., 2005). Manual scaling of the data can be performed by editing the
ionograms using the SAO Explorer software (Galkin et al., 2008). The interferometry
system used by the Digisondes receiver is comprised of four small spaced antennas for





signal reception arranged in a triangle with one antenna at the center. The signals from each
antenna are Fourier analyzed to identify echoes with different Doppler frequencies (for more
details see Reinisch et al., 2004). The Drift Explorer software determines the location of the
source regions of the spread-F echoes for each Doppler component. The ionograms present a
color code showing the direction of echoes that form the spread-F. The sky map and drift
data collected after the ionogram are derived from the measured Doppler frequency and
angle of arrival of reflected echoes. Special processing software enables us to plot skymaps
showing the location of all reflection sources. The Drift Explorer also provides plots of the
drift velocities (zonal, vertical, and meridional components). For more details about
Digisondes sounding modes and drift measurements see Reinisch et al. (2005) and
references therein.

**2.2 Wide-Angle Imaging System**
The airglow images of the OI 630-nm emission used in this study were measured by a
Portable Ionospheric Camera and Small-Scale Observatory (PICASSO) wide-angle imaging
system deployed at Cajazeiras (CZ: 6.87° S, 38.56° W, dip angle: -21.4°), located about
~352 km from south of FZ. It is a miniaturized imaging system that measures the 630.0-nm
and 777.4-nm nightglow emissions. Since the 777.4-nm emission is generally very weak
during solar minimum conditions, we use only the 630.0-nm emission image data for this
study. The PICASSO images are captured on a 1024 × 1024 Andor DU434 CCD with a
spatial resolution of approximately 1 km (azimuthal) over the entire field of view. The
spatial resolution in the radial direction varies from ~1 km to ~5 km from zenith to the edge
of the field of view. The noise contributions from dark current are reduced by cooling the
CCD to at least -60°C. The exposure time for each image is 90 s, and dark images are taken





frequently to remove noise and read-out biases. For details about the data processing from a
similar PICASSO installation, see Makela and Miller (2008).
**2.3 Fabry-Perot Interferometer (FPI)**

FPIs are optical instruments which measure the spectral line shape of the 630.0 nm emission
at around 250 km of altitude and are very useful to study thermospheric winds from Doppler
shifts in the emission's frequency.  For more details of the FPI technique, see Fisher et al.,
(2015) and references therein. The investigation of the departures of the background wind
system can be useful to explain possible sources of the F-uplifts associated with late time RT
instability. For this purpose, we analyzed the behavior of the neutral winds over the
equatorial region taken from a ground-based FPI installed in CZ.

**3  Observations**

**3.1  Spread-F, F-layer height and plasma densities**

We present a case study of a spread-F event which occurred in the June solstice of 2011
during a geomagnetically quiet ($\Sigma Kp = 11$) night an low solar activity with mean F10.7 = 97
SFU (SFU is Solar Flux Unit = $10^{-22}$ W.m$^{-2}$.Hz$^{-1}$).  Fig. 1 shows a sequence of ionograms on
26 July 2011 from 00:40 LT to 03:10 LT over SL (top panel) and over FZ low latitude site
(bottom panel) from 25 July 2011 at 22:00 LT to 26 July 2011 01:30 LT in which the
presence of unusual spread-F patterns is observed. Over SL, the first spread-F trace appears
at 01:00 LT at an oblique angle close to or above the F-layer peak at a virtual range of 600
km. Over the next hour, this structure gradually moves closer to the station SL, finally
merging with F-layer bottom side echoes and becoming a well-formed spread-F trace.

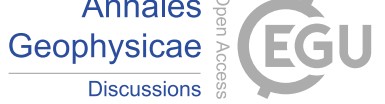

During the spread-F development, it is possible to observe an apparent small increase in the
F-layer heights. Finally, at the end of the spread-F event, around 02:50 LT, we observe a
decreased foF2 and the formation of an Es layer, lasting until 03:10 LT (not shown). We
notice a very similar evolutionary pattern of the structures in the FZ ionograms as in those
obtained from SL. However, the first spread-F traces appeared over FZ around 22:20 LT,
much earlier than over the equatorial site, SL.  These echoes from FZ lasted for about 3
hours. The spread-F echoes gradually move closer to the station or downward to form a
well-structured spread-F pattern.
An important point to be considered is the local ionospheric background in which the
spread-F occurred.  The F-layer parameters, h'F (virtual height of the F-layer bottom side, in
km), the hmF2 (the real height of the F-layer peak, in km) and foF2 (the F-layer critical
frequency, in MHz) for both stations are shown in Fig. 2 from 18:00 LT to 06:00 LT. Over
SL an uplift of the F-layer was observed between 21:00 LT and 23:00 LT, not associated
with any spread-F echoes. Near 21:00 LT we may note some wave-like oscillations in the F-
layer height (notable in hmF2) (with a period on the order of one hour. The first spread-F
trace at oblique angles (and perhaps above the F2 peak) appeared during these oscillations.
On the other hand, over FZ where heights are lower, we observe stronger wave-like
oscillations in both h'F and hmF2 three hours earlier than observed at SL. The F-layer
critical frequency decreased for both stations, as it is expected for this period. At the
beginning of the spread-F occurrence, the foF2 was as low as 4 MHz, corresponding to an
electron density of $1.98 \times 10^5$ el.cm$^{-3}$. The parameter fxI (not shown), or top frequency of
spread-F, which is the highest frequency of spread-F echoes, reached values not higher than
4.5 MHz over SL but reached values around 6.0 MHz over FZ, which means higher plasma
density at this region. Moreover, after the spread-F ceased, it was possible to observe the
recovery of the plasma frequency/density over FZ sooner than over SL.





### 3.2 Depletions in the airglow OI 630.0-nm emission


Figure 3 shows a sequence of four images of the OI 630.0-nm emission collected on  25-26
July 2011 at Cajazeiras (CZ: center of the frame), Brazil. The images are projected over a
geographic map of Brazil assuming an emission altitude of 250 km. The sites of FZ and SL
are also indicated in the top-left panel. Between 23:12 LT to 01:26 LT at least two
depletions can be observed propagating westward. These depletions passed over FZ and CZ
at 23:12 LT, in agreement with the spread-F traces seen in the ionograms from FZ.

### 3.3 F-layer irregularity Drifts – Directions and Velocities

Automatic drift mode routines were used to obtain information about the location of echo
sources in the F-layer associated with plasma irregularities. These routines provide
information about the distance of the reflected echoes, using measurements of the radar
ranges to the vertical and oblique echoes as well as their directions, as described by Reinisch
et al. (2004). The distribution of the echoes can be displayed in skymaps as shown in Fig. 4.
Skymaps between 00:12 LT and 00:42 LT were constructed using data from FZ during the
spread-F event studied where reflected echoes appear and are distributed in a west-east
elongated pattern covering a total horizontal distance of 1200 km (from west to east). It may
be noted that, in general, negative Doppler velocity (yellow color) of the echoes dominates
the western azimuth while the eastern azimuth is dominated by positive Doppler velocity
(blue color), a characteristic that is indicative of an overall westward motion of the
irregularity structures. Additional directional information is obtained from the temporal
evolution of each spread-F echo in plots of the horizontal distance of the echoes (horizontal



axis) as a function of time (vertical axis), presented as directograms. A directogram for the
night of 25-26 July 2011 constructed using data from FZ is shown in Fig. 5. Each horizontal
line of the directogram corresponds to a single ionogram. The spread echoes are distributed
east to west from 21:00 LT to ~ 05:00 LT, although there is only a sparse distribution
between 21:00 and 23:00 LT. The color codes at both sides indicate the incoming and
outgoing direction of the reflectors (irregularities), while the arrow indicates the direction of
propagation. For example, some echoes are seen at ~415-km east around 23:30 LT. The
color code indicates they are at east of the station coming from the east side. Also, there are
echoes at the east of the station which come from northeast, NNE, direction.  Among these
echoes, there are only a few points that are going eastward (blue points).  From 23:30 to
01:30 LT, there are echoes at west which gradually disappear after 02:00 LT. The color code
to the left shows that they are at west from the station and going westward. Thus, the echoes
present a mean westward propagation. We point out that the horizontal distance range limit
is around 600 km, which correspond to an antenna beam angle of approximately 45°, as it is
seen in the directograms on Fig. 5, and hmin is the spread-F reflection height.
The unusual spread-F echoes were observed at both equatorial sites, SL and FZ, with a zonal
separation of ~600 km.  The first spread-F trace was observed at 22:20 LT over FZ and later
at 01:00 LT over SL.  This lag of ~ 02:40 hours suggests an average westward drift velocity
component of ~ 62 ms$^{-1}$. The DPS-4 drift mode provides the full-vector Doppler velocity for
the observed echoes. Figure 6 shows the variation of the $V_z$ (vertical component) and $V_{east}$
(zonal component) velocities taken from measurements of the Digisonde DPS-4 (drift mode)
from 21:00 LT on July 25, 2011, to 04:00 LT on July 26, 2011.  Positive (negative) $V_{east}$
velocities represent eastward (westward) propagation. |V| represents the zonal drift Doppler
velocities are less than 50 ms$^{-1}$, while the maximum vertical upward component is ~40 ms$^{-1}$.
The zonal velocities inferred from Drift Explorer agree well with the estimate obtained from





the difference in onset times of spread-F echoes between SL and FZ, with a mean value of
~55 ms$^{-1}$ during the event. The middle panel is the vector diagram with the variations of the
mean total electrodynamical drift velocity (see Balan et al., 1992). For clarity, the vector
length is fixed, and the information on |V| is represented by the circles (arrow start point).
As it is observed, the vector is found to rotate anticlockwise, starting in the east-up sector in
the night and reaching west-up sector in post-midnight. Velocities extracted from the
airglow images obtained from CZ are shown in the bottom panel of Fig. 6. To estimate the
velocity of the depletion structure, the individual images were processed by first spatially
registering the 630.0-nm images using the star field. After removing the stars from the
images using a point suppression methodology, the images were projected onto geographic
coordinates assuming an airglow emission altitude of 250 km (for details of analysis
technique see Chapagain et al., 2012). The depletion structure was selected in consecutive
images to find the zonal shift of the structures from which the velocity was estimated. The
estimated zonal propagation velocity was ~60 ms$^{-1}$, which agrees well with the velocities
determined by the Doppler technique of the Digisonde. We should keep in mind that the
Digisonde Doppler technique determines the mean irregularity motion while the velocities
from the airglow technique estimate the mean propagation of the plasma depletion.

299        Figure 7 presents the variation of F-layer height (fixed frequencies) in both stations,

SL and FZ. This plot can be useful to analyze the oscillations in the F-layer bottom side and
the possible association with gravity waves.

**3.4 Thermospheric Winds**
Figure 8 shows the measured thermospheric zonal (top panel) and meridional (bottom panel)
wind on July 25-26, taken from the FPI installed in CZ, the same location where the airglow
images were obtained. The shaded region is the standard deviation of the monthly average,





the green lines are the average winds on July 25-26 (±2days), and the red line is the
measurement for July 25-26. It is observed that on July 25-26 between 22:00 LT and 01:00
LT the zonal wind is abnormally eastward (~100 m/s), while the meridional wind departs
from the monthly and daily variation average. Additionally, it is observed an equatorward
wind (~30 m/s). From this, we can consider that a possible balance between the zonal and
meridional wind component may be responsible by plasma advection (plasma movement)
from low latitude to equatorial region, which might have maintained the F-layer at a higher
altitude as discussed by Nicolls et al., 2006. This apparent uplifts observed in both stations
around 00:00 LT might have caused the development of late RT-instability and the PMIs.

**4 Discussion**

We present an unusual event of PMIs/spread-F/depletions over the equatorial site in Brazil
that exhibits singular features. This is the first report of such distinct type of spread-F for the
Brazilian equatorial region, though it was observed earlier at the low latitude station CP
(Brazil) for the solar minimum 2008-2009 by Candido et al., 2011. A careful analysis of
equatorial ionograms and other plots from digisonde soundings suggest modifications in the
ionospheric plasma density structuring, such as those associated with plasma density
depletions, which are responsible for a variety of spread F-layer patterns.

**4.1 Depletions in the airglow OI 630.0 nm images**

Airglow images show an apparent southwestward propagation of depletions on this night,
which differs from the typical propagation direction of post-sunsets EPBs. However, this
atypical propagation can be a characteristic of post-midnight depletions and needs further



investigation with a long-term airglow database. The depletions also propagated over CZ
(350 km south of FZ) with mean westward velocities ~60 ms⁻¹ which are similar to the
velocities of propagation of the irregularities observed with the Digisonde at FZ. Some
authors have demonstrated that EPBs can also present westward propagation after midnight
during quiet times (Paulino et al., 2010; Sobral et al., 2011). However, they defined in those
studies that the depletions associated with EPBs should first present movement to the east
earlier in the evening and reversal to westward at later hours. This is not the case for the
structures presented in this work since there are no depletions in the OI 630.0-nm images
propagating eastward earlier in the evening.
Moreover, Sobral et al. (2011), interpreted that westward traveling plasma bubbles (WTPB)
observed at the same region were associated with westward zonal thermospheric winds
(simulated results). On the other hand, Fisher et al. (2015) presented a climatological study
of the quiet time thermospheric winds and temperatures by measurements of the OI 630.0
nm airglow emission spectral line shape over the same region.  They noticed that during low
solar activity (F10.7 < 125 sfu), the zonal and meridional winds are, on average, negligible
in postmidnight hours. It is possible that these differences can be attributed to departures
from the wind system at which could be responsible by the F-layer uplifts and plasma
instabilities/irregularities development.

**4.2 Spread-F in ionograms**
As mentioned before, spread-F echoes in ionograms generally appear first at the low-
frequency end, as satellite traces, evolving into spread-F echoes extended in frequency and
range. These characteristics were not seen in the present study. In this work, the reflected
echoes observed in the ionograms first came from oblique directions and at heights which
could be considered possibly higher than those observed overhead. The spread echoes





appear at the higher frequency edge of the F-layer, with top-frequency higher than the layer
critical frequency. Subsequently, the low-frequency edge of the cusp merges with the main
trace, while the baseline of the spread-F traces gradually decreases in height. Anomalous
traces in F-layer ionograms, such as 'cusps' or 'spurs,' were described in earlier studies to
be associated with traveling disturbances in the ionosphere. Munro and Heisler (1956) and
Heisler (1958) have observed the occurrence of anomalous traces in ionograms and
attributed them to the manifestations of TIDs. As it is well known, TIDs can be described as
frontal gravity waves propagating horizontally in the ionosphere, causing increases and
decreases in the ionization, i.e., horizontal gradients in the ionization. According to Munro
and Heisler (1956), changes in the ionization would be responsible for the anomalous traces
in the F-layer ionogram. Similar occurrences were reported by Ratcliffe (1951) for
ionograms from Huancayo, Peru. Calvert and Cohen (1961) have pointed out that some
spread-F traces observed over Huancayo presented characteristics similar to frequency
spread-F from "temperate" latitudes, which are mainly associated with TIDs. Also, they
studied distinct configurations of spread-F with echoes coming from oblique directions,
similar to what is presented in this work. The oblique echoes observed in ionograms alone
could not provide their zonal direction (from east or west). However, additional directional
information provided from the drift mode sounding of the Digisonde DPS-4 and their
appearance first in the ionograms over FZ followed by their occurrence over SL (a western
site in relation to FZ), suggested that they propagated westward.  Late/pre-dawn spread-F
was also reported by McDougall et al. (1998) for solstices in the Brazilian sector. However,
they considered the occurrence of late time spread-F during December solstice at Fortaleza
as patches of ionization, which cause spread echoes at the high-frequency end or the
frequency spread-F. They also concluded that the echoes did not come from overhead
structures but from the east or west directions.



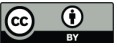

**4.3 Post-midnight irregularities/F-region background conditions**

As it is well-known, the poor alignment between the sunset terminator and the magnetic
field lines during June solstice in Brazil is responsible by the low occurrence rate of post-
sunset spread-F/EPBs, since the vertical plasma drifts are very weak. However, it is
observed a secondary occurrence peak of spread-F/plasma irregularities in June solstice
during late night, especially in post-midnight. For this, it is necessary to have an F-layer
uplift, which creates favorable conditions for the development of the RT instability. These
conditions are not completely understood, and they have been discussing by several authors
(McDougall et al., 1998; Nicolls et al., 2006; Abdu et al., 2009; Nishioka et al., 2012,
Yokohama et al., 2012, Ajith et al., 2016).
During the high solar activity, the longitudinal variation of the declination angle is
predominant on the F-layer vertical drift and the occurrence of the plasma irregularities,
while it is not important during solar minimum. During low solar activity/solar minimum, in
the absence of geomagnetic disturbances, the seeding processes related to gravity waves
seem to be more important, especially when the PRE-amplitude is small or absent
(Balachandran et al., 1992; Abdu et al., 2009). In this way, we should address the conditions
which precede the occurrence of the post-midnight irregularities observed in this work. It is
noticed that spread-F traces associated with plasma irregularities were detected firstly at
oblique directions at least 500 km at east or west from the station, as seen in the
directograms in Figure 5, which we can consider as ionospheric conditions favorable in a
wide longitudinal range.





### 4.3.1 Thermospheric winds



Nicolls et al. (2006) discussed the nocturnal F-layer uplifts associated with the
secondary maximum of spread-F occurrence rate in low solar activity. As it is well
understood, the nocturnal westward electric field is responsible by the downward movement
of the F-layer. During solar minimum, these electric fields can be easily reversed by a weak
geomagnetic disturbance. However, in the absence of the geomagnetic disturbance, which is
the case studied in this work, other sources should be considered. Analyzing F-layer uplifts
for different conditions of solar activity, Nicolls et al. (2006) verified that during downward
F-layer movement (decreasing westward electric field), even a small contribution of a
meridional equatorward wind (~30m/s), could lead the F-layer to higher heights triggering
the RT instability.
Moreover, it was discussed that neutral winds could not uplift the equatorial plasma directly,
but they are sources of meridional advection (movement) of plasma, driven by a latitudinal
gradient in electron density, responsible by F-layer uplifts. They concluded that the uplifts
could be due to the decreasing, not to the reversal, of the westward zonal electric field
associated with departures in the wind system related to the midnight temperature maximum
(MTM), recombination processes, and the plasma flux. In this way, we analyze the zonal
and the meridional neutral wind variation in Figure 8, in order to verify that there are
suitable conditions for F-layer uplift. As it is observed in Figure 8 (top panel), the zonal
wind is ~100 m/s just before midnight while meridional wind (equatorward) is ~30 m/s just
after midnight (bottom panel). There is evidence that the mean equatorward meridional
winds have kept the F-layer at higher altitudes enough to the trigger the RT instability
development.






### 4.3.2 Recombination processes - Rayleigh Taylor instability growth rate

Nishioka et al. (2012) discussed the causes of the postmidnight uplifts that occurred during
winter in Chumphon, Thailand (low latitude) and the post-midnight Field-Aligned
Irregularities, FAIs, in Kototaband, Indonesia (equatorial region). As it is well known, the
zonal electric field is westward during the night, as the vertical drift **ExB** is downward. This
condition leads to an RT-instability growth rate negative. In this way, it is important to
address, the importance of the term $g/\nu_{in}$ in the linear growth rate of RT-instability, and of
the recombination processes, as shown in Equation (1):
$$\gamma = (\frac{E}{B} + \frac{g}{\nu_{in}})\frac{1}{L} \qquad (1)$$

Where: E is an the electric field; B is the magnetic field; g is gravity acceleration, $\nu_{in}$ is ion-
neutral collision frequency; L is the scale length of the vertical gradient of the F-region
plasma density. At night, the zonal electric field is westward, as the growth rate can be
negative, i.e., the F-layer bottom side is stable. On the other hand, the term $g/\nu_{in}$ may hands
out in the following conditions: 1) $\nu_{in}$ is proportional to the neutral density, *n*, where *n* is
smaller during the night than the day; 2) $\nu_{in}$ is smaller at higher altitudes owing to the
decrease of *n* with the height; 3) $\nu_{in}$ is smaller during low solar activities. Therefore, under
the appropriate conditions, the RT growth rate can be positive, although small, as it is
observed in this work. To understand the recombination processes as a source of the F-layer
uplift it should be considered that the F-layer bottom side is eroded if it is at lower altitudes
(at ~300 km), such as there is a decreasing of peak density and the increasing of F-layer
peak height. For clarity, we present the F-layer density profiles, in Fig. 9, taken from
measurements using the Digisonde installed in SL. It is possible to observe that from 22:00
to 00:00 LT, the F-layer peak height, and peak density decrease. As the F-layer bottom side





is at a lower height, it is observed an apparent F-layer uplift, which can be attributed to the
recombination process at the bottom side.

**4.3.3 Es-layer electric fields**

The role of Es-layer has been considered as a possible cause for the late-time RT instability
development. Low latitude Es-layer can provide enough polarization electric field which
maps to equatorial F-layer bottom side, causing F-layer uplift, as pointed out by Yizengaw
et al. (2013). They interpreted the occurrence of late plasma irregularities/EPBs over Africa
coast during the same period of this work, June solstice 2011, and discussed that during
quiet geomagnetic nights, there were favorable conditions for the action of polarization
electric fields associated with low latitude Es layer/instability which mapped to the
equatorial F-layer along the geomagnetic field lines seeding RT-instability and irregularities.
In fact, in this work, we can observe the occurrence of the Es-layer at the both quasi-
equatorial station FZ and at SL, at around 00:00 and 02:50 LT respectively. However, the
influence of Es-layers on late time F-layer uplift in this work is not clear since they occur at
the same location of the spread-F. Its influence on the post-midnight spread-F during solar
minimum is worth of investigation in further works.

**4.3.4 Mesoscale Travelling Ionospheric Disturbances, MSTIDs and Gravity-Waves,**
**GW**
MSTIDs have been reported in Brazilian low latitudes using airglow and ionosonde
(Candido et al., 2008, 2011; Pimenta et al., 2008). They appear as large-scale dark bands
aligned from northeast to southwest propagating northwestward mainly during low solar
activity and are associated with electrodynamics forces in mid-latitudes (Perkins instability)



or by the propagation of gravity waves in ionospheric heights at low latitudes or equatorial
region. If they propagate at equatorial ionospheric heights, they can be seen as oscillations in
the F-layer bottom side and can trigger RT-instability and plasma bubbles. In this work, the
plasma irregularities seen by the ionosonde are preceded by small oscillations in the F-layer
bottom (h'F) and peak heights (hmF2). However, oscillations are usually observed in the F-
layer bottom side, and it should be carefully considered in order to establish if they are
associated with GWs. Generally, they are considered associated with GW if it downward
phase propagation is observed in the fixed frequencies (isolines) plots, i.e., the oscillations
are seen firstly in the higher frequencies. Figure 7 showed the occurrence of oscillations in
F-layer through some fixed frequencies (isolines) in both stations FZ and SL, although the
downward propagation is not exactly clear. On the other hand, the spread-F pattern observed
in this work is quite similar to those reported by Candido et al. (2011) during the descending
phase/solar minimum at low latitudes in CP. This feature could suggest that they could be
caused by low latitudes MSTIDs propagating equatorward or associated to the action of
polarization electric fields mapping from low latitudes MSTIDs structures to the equatorial
F-layer bottom side. This kind of event was reported by Miller et al. (2012), which studied
the occurrence of EPBs on the same night of the occurrence of MSTIDs propagating in mid-
latitudes and attributed them to the action of the electric field from these MSTIDs in the F-
layer region. However, the depletions observed in the OI 630-nm emission (Figure 3)
present distinct features (propagation direction) of those associated to MSTIDs coming from
low latitudes reported by Candido et al. (2011). Also, they are not similar to the depletions
associated with the typical EPBs which propagate eastward. Recent results by Takahashi et
al. (2018) reported the occurrence of equatorial MSTIDs in high solar activity conditions
(2014/15), which were associated with periodic plasma bubbles in the Total Electron





Content (TEC) maps in the same region. They showed evidence of tropospheric sources for
the development and propagation of GWs at ionospheric heights.
Finally, we should address that, as shown in Figs.2 and 7, late height rise (in both h'F and
hmF2) with smaller amplitude waves are observed at SL starting at ~ 21:00 LT when the
base height (h'F) increased to > 250 km. Such a condition can be suitable for the growth of
RT instability. Over FZ, a similar sequence of variations occurred starting at ~23:00 LT in
hmF2. Notice that h'F and hmF2 values were significantly smaller than those at SL.
However, it is notable that the oscillations in the F layer heights, especially in hmF2, (with
the period around 36 min) that preceded the spread F traces (at both sites) are significantly
higher in amplitude at FZ than at SL. This aspect can be noted in more detail in the iso-line
plots of plasma frequencies presented in Fig. 7, where in the height oscillations show larger
amplitude and occurring at earlier local times than they are at SL. Such oscillations may be
associated with gravity waves propagating to ionospheric heights with preferential
propagating directions to northeast and southeast, as recently reported by Paulino et al.
(2016). These oscillations are indicative of the seed perturbations to lead to the SF
irregularity development through RT mechanism. Depending upon the amplitude of the seed
perturbation, even the small increases in the F layer height that marked this period, could be
capable of seeding RT instability and consequently generate the spread F irregularities (see,
for example, Abdu et al., 2009). To explain the non-local origin of the SF traces, as observed
at both sites, it will be necessary to assume that the precursor conditions that existed at SL
and FZ must have continued to exist in longitude extending further eastward of Fortaleza,
perhaps with some increase in intensity so that the irregularities generated therein and
drifting westward could be the origin of the oblique spread F trace first observed over FZ
and later over SL.





It is plausible to consider that the depletions observed in this work can be associated with
atypical EPBs triggered by GWs/MSTIDs at locations at the east of FZ and SL or to F-layer
uplifts caused by departures from wind system simultaneously to a weakening of the
westward zonal electric field (not shown here) during low solar activity. We should notice
that the observational techniques used in this work are complementary and validate each
other    to    identify    "anomalous"    spread-F    patterns    associated    with    plasma
irregularities/depletions and can help the understanding of the ionosphere during low solar
activity. The drift mode is very useful and suitable for tracking plasma irregularities and
their evolution in the absence of other techniques.

**5  Summary and Conclusions**
In this paper, we have presented and discussed an unusual spread-F pattern associated with
unusual depletions on the OI 630.nm airglow emission observed during geomagnetically
quiet conditions during the June solstice of 2011 over the equatorial region in Brazil. We
summarize our findings as:
1) The unusual spread-F pattern studied in this work present a distinct feature from those
usually observed at post-sunset hours;
2) The spread-F/depletions occurred during low plasma density conditions, geomagnetically
quiet nights, low solar activity and propagated westward.
3) The processes to generate spread-F at equatorial latitudes during quiet time seems to be
associated with the late time F-layer uplifts, possibly caused by departures in the neutral
wind system, probably associated with a weakening of the westward electric field, or to the
propagation of GWs at ionospheric heights, which favor the development of the late-time
RT-instability.



4) The spread-F event discussed here presents characteristics similar to those of the earlier
cases reported for low latitudes in CP during June solstice of solar minimum 2008-2009 by
Candido et al., 2011.

The instrumental approach in this work seems to be suitable for further ionospheric studies,
modeling, and forecast during low solar activity.


**Abbreviations**

SL: Sao Luis
FZ: Fortaleza
CZ: Cajazeira
CP: Cachoeira Paulista
DPS: Digital portable sounder
FPI: Fabry-Perot Interferometer
EPBs: Equatorial plasma bubbles
PMIs: Postmidnight plasma irregularities
RT: Rayleigh-Taylor
EIA: Equatorial ionization anomaly
MSTIDs: Meso-scale traveling ionospheric disturbances
FAIs: Field-aligned irregularities
TIDs: Travelling ionospheric disturbances
SFU: solar flux unity
LT: local time
UT: Universal time





MTM: midnight temperature maximum
GWs: Gravity waves

**Data availability**
The processed data used in this work can be requested to the author CMNC by the email:
claudia.candido@inpe.br. The raw data: Digisonde data is available in the website:
www.inpe.br/embrace. The airglow and Fabry-Perot data should be requested to the author:
JM, by the email: jmakela@illinois.edu.

**Author Contributions**
CMNC wrote the manuscript and plotted the graphics of the ionospheric parameters. FBG
helped with part of the graphics and revised the manuscript. JS, ISB, EC, MAA, N.B., ZL,
CW read and made suggestions to the manuscript. JM and NC provided the airglow figures
and Fabry-Perot data and plots, as well as read the manuscript and suggested corrections. All
the authors read, give comments and suggestions to the work and agree with the content and
submission of this manuscript.

**Competing interests**
The authors declare they have no conflicts of interest.

**Acknowledgments**
C.M.N.C thanks the Brazilian funding agency CNPq for the financial support through the
process n.64537/2015-5, and to China-Brazil Joint Laboratory for Space Weather and the
National Natural Science Foundation of China for the project with No.41474137 and
1674145 for the postdoctoral fellowship. Also, we thank INPE technical staff for the
assistance with the instrumentation and data management. N.P.C. was supported by the
NASA Living With a Star Heliophysics Postdoctoral Fellowship Program, administered by





the University Corporation for Atmospheric Research (UCAR). Work at the University of
Illinois at Urbana-Champaign was supported by National Science Foundation CEDAR grant
AGS 09-40253 and was performed in collaboration with J. W. Meriwether at Clemson
University. We are grateful to the Universidade Federal de Campina Grande and Dr. Ricardo
A. Buriti for the support to the imaging systems installed at Cajazeiras.

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





Figure 1

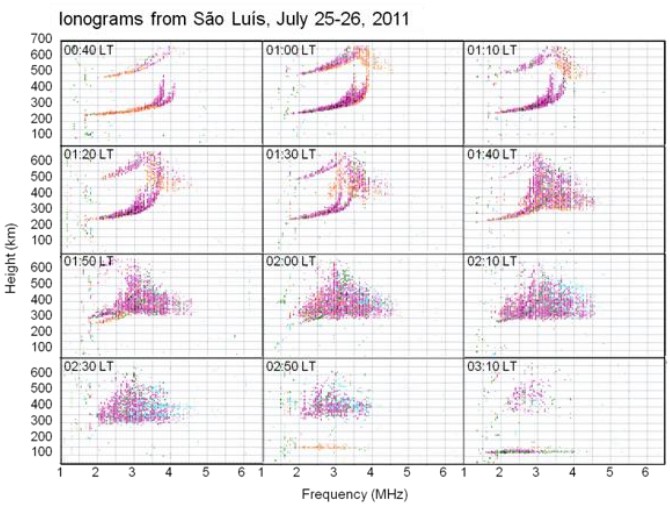

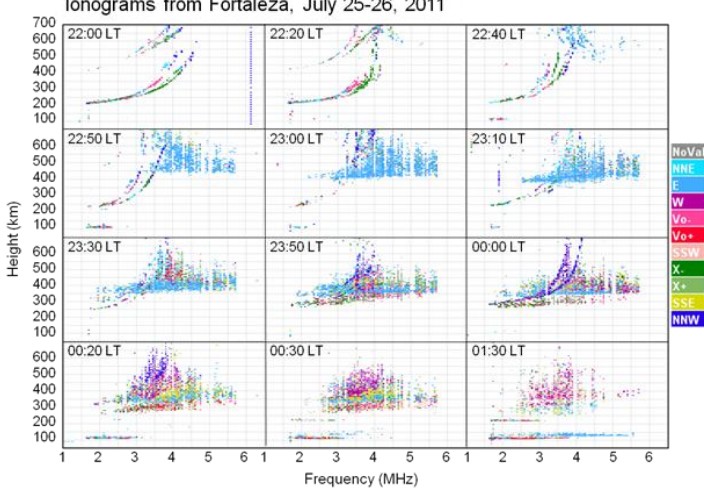



**Figure 1**: Sequence of ionograms obtained on July 25-26, at São Luis, from 00:40 to 03:10
LT and over FZ, Brazil, 2011, from 22:00 to 01:30 LT. The spread-F shows an unusual
pattern, with oblique echoes. The color scale in FZ ionograms indicates echoes are coming
from the east and propagating to the westward.



Figure 2

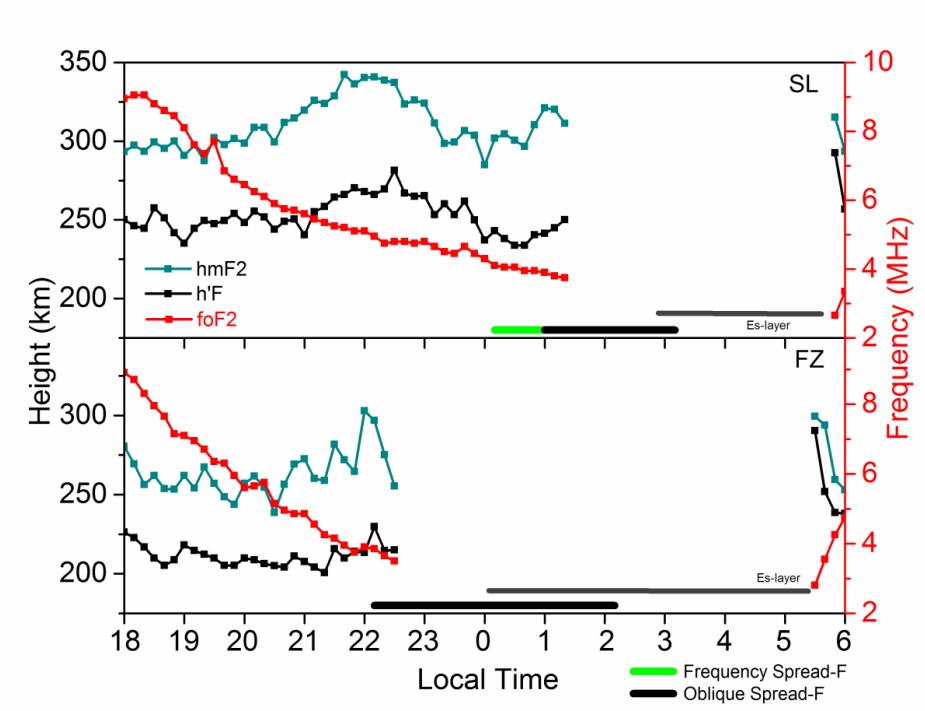


**Figure 2**: F-layer parameters h'F (km), hmF2 (km) and foF2 (MHz), on July 25-26, 2011
obtained from the Digisondes at São Luis and Fortaleza.













Figure 3

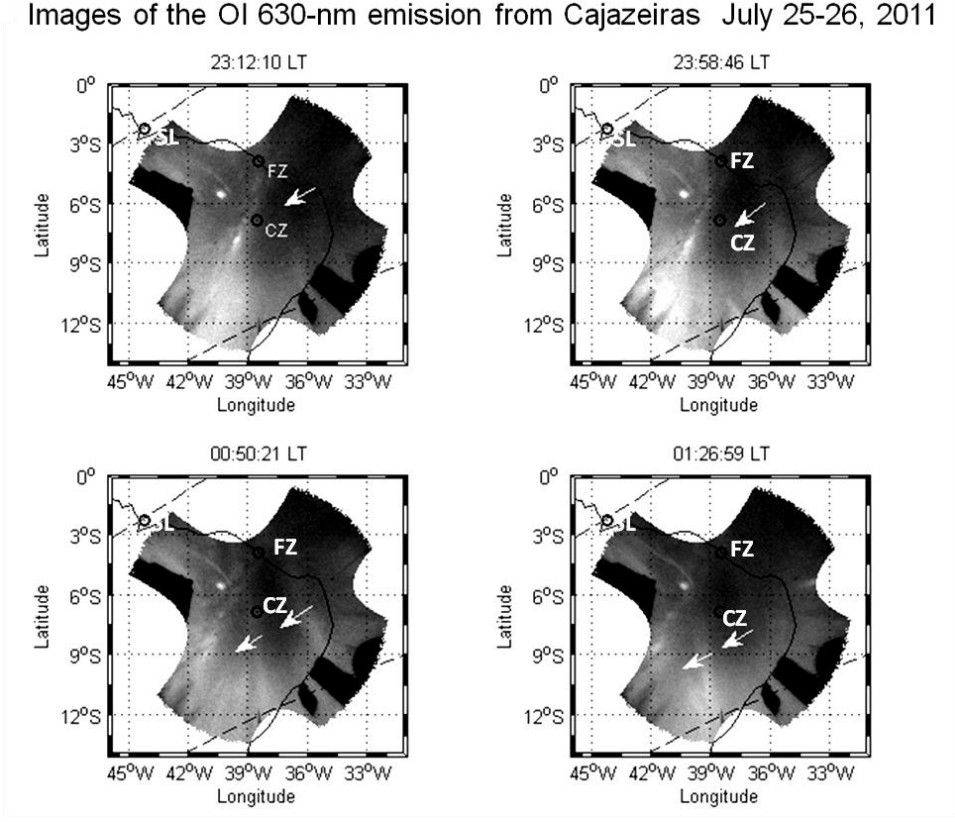


**Figure 3**: Sequence of OI 630-nm images showing the time evolution of depletions on July
25-26, 2011, between 23:12 LT and 01:26 LT at Cajazeiras, Brazil. The images are
projected onto geographic coordinates over the Brazil map. In the plot, FZ is Fortaleza, SL
is Sao Luis, and CZ is Cajazeiras. Arrows indicate the propagation direction of the
depletions.




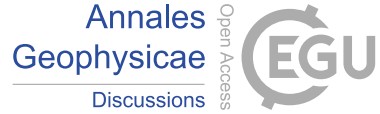

Figure 4

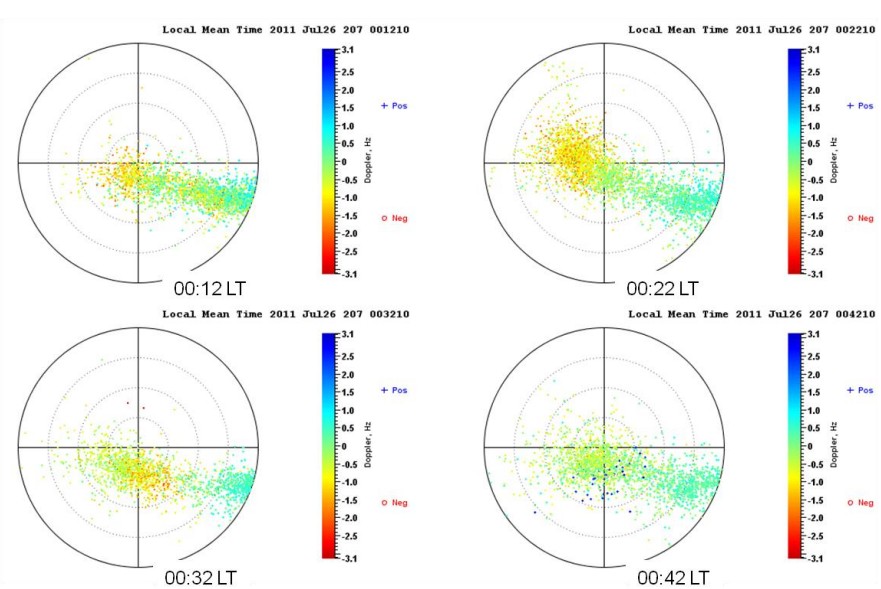



**Figure 4**: Skymaps registered over FZ from 00:12 LT to 00:42 LT on July 26, 2011,
showing the echoes location and Doppler frequencies (color-coded) for F-region
echoes from Digisondes. Doppler velocities: Positive: irregularities arriving at the
station; Negative: irregularities leaving the station.












Figure 5

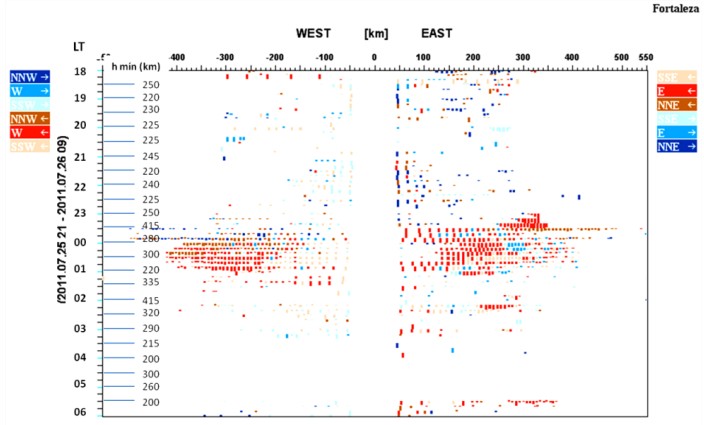


**Figure 5:** Directogram for Fortaleza on July 26 showing the location and the horizontal
distances of the irregularities detected by Digisonde and seen in the ionograms as spread-F.
At left:  F-region height (km), where hmin is spread-F reflection height.



















Figure 6

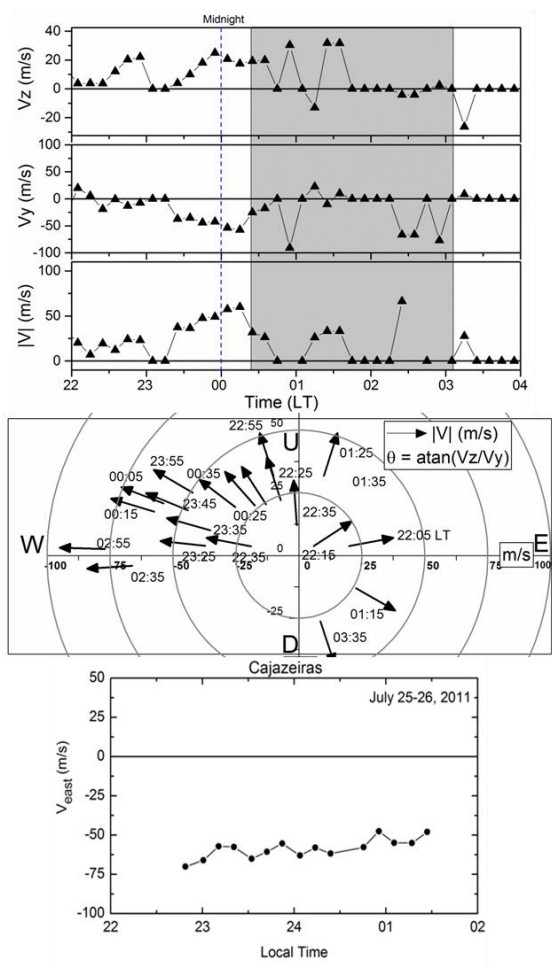


**Figure 6**: Top Panel: Vertical (Vz) and zonal drift (Vy) velocities on July 25-26, 2011 over
FZ from 22:00 LT to 04:00 LT. $V_{east} > 0$. Middle Panel: Vector diagram showing the
variations and directions of the mean total drift velocity of the irregularities seen as spread-F
in ionograms. For clarity, the |V| values are represented by the arrow start points. Bottom

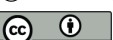



panel: Zonal drift velocities obtained from the depletions seen on the OI 630.0 nm emission
images obtained at CZ on July 25-26, 2011 for comparison.


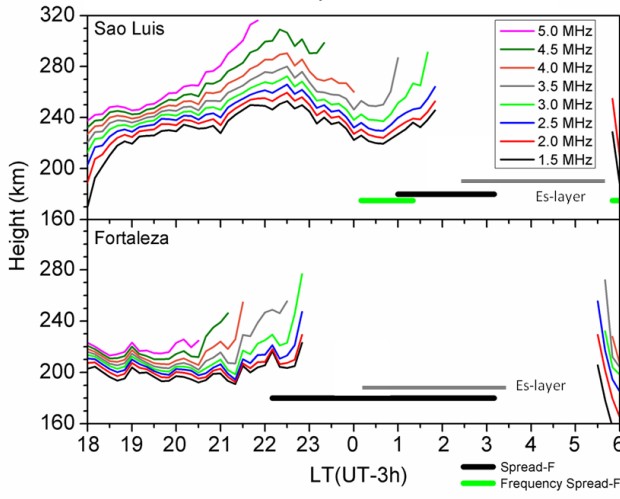


**Figure 7**: Oscillations in the real height of F-layer, at fixed frequencies during the spread-F
in São Luis (top panel) and Fortaleza (bottom panel).















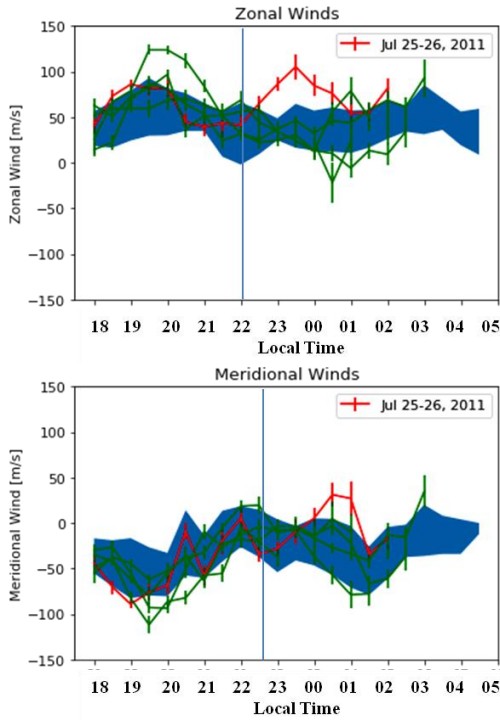


Figure 8: Measured Zonal and Meridional Winds in CZ, Brazil, in July 2011. The shaded
region is the monthly average, the green lines are the mean winds on July 25-26 (mean of 2
days), and the red line is for July 25-26.








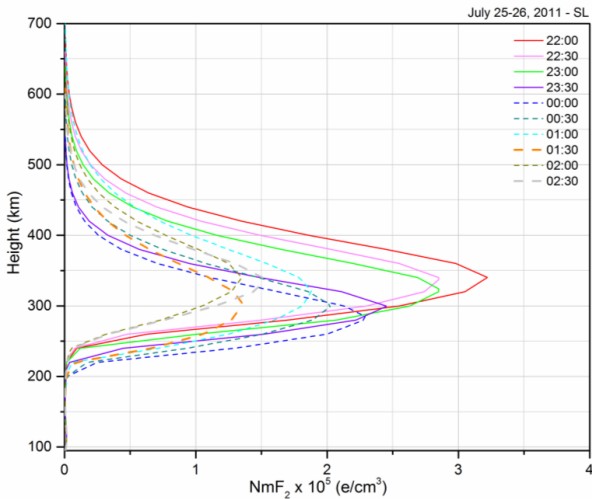


Figure 9: F-layer plasma density profile for July 25-26, taken from Digisonde measurements
installed in SL, and by Sao-Explore inversion techniques.
















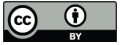

**List of figures**

Figure 1: Sequence of ionograms obtained on July 25-26, at São Luis, from 00:40 to 03:10
LT and over FZ, Brazil, 2011, from 22:00 to 01:30 LT. The spread-F shows an unusual
pattern, with oblique echoes. The color scale in FZ ionograms indicates echoes are coming
from the east and propagating to the westward.

Figure 2: F-layer parameters h'F (km), hmF2 (km) and foF2 (MHz), on July 25-26, 2011
obtained from the Digisondes at São Luis and Fortaleza.

Figure 3: Sequence of OI 630-nm images showing the time evolution of depletions on July
25-26, 2011, between 23:12 LT and 01:26 LT at Cajazeiras, Brazil. The images are
projected onto geographic coordinates over the Brazil map. In the plot, FZ is Fortaleza, SL
is Sao Luis, and CZ is Cajazeiras.

Figure 4: Skymaps registered over FZ from 00:12 LT to 00:42 LT on July 26, 2011,
showing the echoes location and Doppler frequencies (color-coded) for F-region echoes
from Digisondes. Doppler velocities: Positive: irregularities arriving at the station; Negative:
irregularities leaving the station.

Figure 5: Directogram for Fortaleza on July 26 showing the location and the horizontal
distances of the irregularities detected by Digisonde and seen in the ionograms as spread-F.
At left: F-region height.



Figure 6: Top Panel: Vertical (Vz) and zonal drift (Vy) velocities on July 25-26, 2011 over
FZ from 22:00 LT to 04:00 LT. $V_{east} > 0$. Middle Panel: Vector diagram showing the
variations and directions of the mean total drift velocity of the irregularities seen as spread-F
in ionograms. Bottom Panel: Zonal drift velocities obtained from the depletions seen on the
OI 630.0 nm emission images obtained at CZ on July 25-26, 2011.

Figure 7: Oscillations in the real height of F-layer, at fixed frequencies during the spread-F
in São Luis (top panel) and Fortaleza (bottom panel).

Figure 8: Measured Zonal and Meridional Winds in CZ, Brazil, in July 2011. The shaded
region is the monthly average, the green lines are the mean winds on July 25-26 (mean of 2
days), and the red line is for July 25-26.

Figure 9: F-layer plasma density profile for July 25-26.







