# Peer review of "Postmidnight equatorial plasma irregularities on June solstice during low solar activity"

_Annales Geophysicae, 2018_

## Referee Comment (RC1) · Anonymous Referee #1 · 31 Jan 2019

In this paper, the authors reported a case of equatorial F region plasma irregularities appearing at post-midnight over Brazil. The irregularity first appeared around the F layer peak and then evolved into mixed spread-F. Simultaneous airglow observations showed plasma depletions drifting westward. By employing the simultaneous digisonde and FPI measurements, the authors suggested that the generation of midnight F region irregularities could be linked with the uplift of F layer at late night which favor the development of RT instability. I recommend the paper for publication in AG.

Specific comments are as follows, 1. lines 59-61, there have been a lot of papers on June solstitial post-midnight F region irregularities. Although post-sunset F region

irregularities were addressed in the review paper by Balan et al. (EPP, 2018), another review paper by Otsuka (PEPS, 2018, Review of the generation mechanisms of post-midnight irregularities in the equatorial and low-latitude ionosphere) is closely relevant to the topic and should also be referenced here.

2. lines 126-128, the authors mentioned that the post-midnight F region irregularity is unusual compared to post-sunset F region irregularity. However, post-midnight F region irregularities have been widely investigated. Is the present case significantly different from the post-midnight F region irregularities reported earlier?

3. lines 336-340, for the present case, is it possible that the irregularities generated at eastern longitudes of Cajazeiras after sunset when the irregularities drifted eastward slowly, and some time later, the irregularities drifted westward and then were observed by the imager at Cajazeiras? If so, the case would be similar to previous observations.

4. lines 534, Please clarify what is the "anomalous pattern". Westward drifts of F region irregularities at post-midnight during geomagnetic quiet conditions are not rare.

5. lines 544, as mentioned above, what is the distinct feature compared to previous results of post-midnight F region irregularities?

6. lines 548-551, it looks like that all the known factors affecting the generation process of F region irregularities are possible. If so, the conclusion does not make sense. Is it possible to determine the main factor(s) causing the generation of post-midnight irregularities through a quantitative simulation in terms of the ionosonde and FPI observations?

7. Figure 3, it's difficult to find the depletion. Please highlight the structure in the plot.

---

## Author Comment (AC1) · 7 Mar 2019

Comment #1: Answer: Thank you. We agree that the reference by Y. Otsuka (2018) is relevant to the topic. We have included it in the revised paper.

Comment #2 Answer: We understand that the reported spread-F, as seen in the ionograms is unusual with relation to the well-known sunset spread-F, since they appear at the higher frequency edge and progressively evolutes to a mixed (range and frequency) spread-F. Earlier reports showed that PMIs occur during very quiet geomagnetic conditions in June solstice and during low solar flux conditions as it is discussed in the present case. They are commonly reported as as FAIs in the observations taken from

coherent and incoherent radar (Otsuka et al., 2009; Yokohama et al, 2011; Nishioka et al., 2012; Dao et al., 2017; Zhan et al. 2018), as well as are observed as depletions in the plasma density taken from satellite measurements (Dao et al., 2013; Yizengaw et al., 2013) or mild frequency spread-F seen in ionograms taken from off-equator stations. From our knowledge our PMIs observations were made using ionosondes and airglow simultaneously for the first in Brazil in the context of spread-F morphology and evolution.

Comment #3 Answer. Sure. This hypothesis is plausible and it was considered. However, there are no depletions in the OI 630.0 nm images neither spread-F in ionograms earlier in the night.

Comment #4 Answer. The anomalous patterns are related with its first appearance of spread-F echoes at the higher frequency edge of F-layer trace. Generally, the first spread-F echoes can appear as satellite trace at post sunset times or in the lower frequency edge. This anomalous pattern could be addressed as spur traces in the ionograms. Westward drifts are not rare, as mentioned by Otsuka et al., 2008. In Brazil westward depletions were observed during low solar activity and associated with previous depletions drifting eastward by Paulino et al., 2011. It is not the case studied in this work.

Comment #5: Answer. Please, see the answer 4.

Comment #6: Answer. It is currently accepted that a combination of factors can be responsible by the generation of PMIs or of these transient irregularities, especially owing the quiescent ionosphere. I personally agree that simulations could be useful to better investigate PMIs. However, we are performing analysis of other recent cases using the same instrumental approach and the simulation will be considered.

---

## Referee Comment (RC2) · Anonymous Referee #2 · 24 Apr 2019

General comments:

Presented in this manuscript is an analysis of an event in which postmidnight equatorial plasma irregularities occurred over Brazil during the June solstice of 2011. The peculiar feature of this event is the fact that the irregularities drift from east-to-west, as opposed to west-to-east. Such a phenomenon is typical of a geomagnetic storm, in which the disturbance dynamo results in a westward thermospheric wind at the equator that has any equatorial plasma bubbles (and the plasma irregularities therein) imbedded among the neutrals. A good level of observations supports the overall conclusions in this work, including ionosondes, Fabry-Perot Interferometers and all-sky cameras.

[Figure]

The potential role of atmospheric gravity waves and background neutral winds are discussed as potential candidates for the presence of plasma irregularities during this particular evening. It is concluded that departures from the typical thermospheric wind system could be to blame for the increase in R-T growth conditions. The manuscript is mostly very well written and is easy to follow, and the examination and discussion of observations presented in the results section is relevant to the field and is acceptable for publication in Ann. Geophys. in the view of this referee, following the consideration of a few minor comments and suggestions, as detailed below.

Specific comments:

1. Line 37 – F10.7 values need units

2. Lines 59-60 – Please be more specific than "distinct longitudinal sectors"; where do these PMIs occur?

3. Line 123 – again, please specify which longitudinal sectors

4. Lines 221-222 – "appeared during these oscillations" This feature isn't obvious in Fig 2. Can you please elaborate?

5. Lines 238-239 – FZ and SL are indicated in all panels, are they not?

6. Lines 239-240 – These two depletions are particularly difficult to see, particularly for readers unfamiliar/inexperienced in examining all-sky-camera data. Is there a way to make these features more obvious?

7. Figure 4 – Axes on these plots require labelling (i.e., N-E-S-W)

8. Lines 260-261 – there is a rather sparse distribution between 2 and 5 LT as well, is there not? Related to this figure, are the two color panels on the left and right indicate the directions on the left and right of the plot, respectively? This isn't entirely clear.

9. Lines 265-266 – are the echoes in the NNE direction also at 23:30 LT and 415 km? The presence of these echoes is not clear to this reader.

10. Line 280 – the maximum upward velocity appears closer to 30 m/s to this reader.

11. Line 285 – the circles are not clearly visible, have they been removed?

12. Lines 299-301 – more discussion/description is needed for the wave structures in Figure 7. These features are highlighted and discussed later in the discussion section, but they should first be highlighted here.

13. Lines 320-322 – The authors should specify here what makes this event "distinct"; i.e., they should mention their non-customary westward propagation.

14. Figure 8 caption – "The shaded region is the monthly average" is rather misleading. I assume that the shaded region indicates the mean +/- one standard deviation. If this is the case, both the caption and the manuscript text (i.e., lines 306-307) should be clarified.

Technical corrections:

1. Line 52 – "small-scale" and "large-scale"

2. Line 157 – "small-speed"

3. Line 186 – "that" instead of "which"

4. Line 190 – "late-time"

5. Line 199 – "night and low"

6. Line 201 – "low-latitude"

7. Lines 279-280 – "|V| represents the zonal drift Doppler velocities are less than 50 m/s" does not read well. Please reword

8. Lines 310-311 – Sentence beginning with "Additionally" also does not read well. Please reword.

9. Line 321 – "low-latitude"

10. Line 330 – "post-sunset"

11. Lines 386-387 – "However, it is observed a secondary occurrence peak…" does not read well. Please reword.

12. Line 390 – "discussed", not "discussing"

13. Lines 437-438 – "This condition leads to a negative RT-instability growth rate."

14. Lines 445-446 – It isn't clear to this reader what the authors mean by "may hands out". Please rephrase.

15. Line 462 – "Low-latitude"

16. Several references listed in the references section do not appear within the manuscript (e.g., Abdu et al., 1981b; Abdu et al., 1982; Bastia et al., 2004; and Carter et al., 2013, there could be others). The authors are encouraged to make sure that each paper listed has been cited at an appropriate location within the manuscript text. Also, Dao et al., 2016 appears in the text as "Dao et al., 2017".

---

## Author Comment (AC2) · 21 May 2019

Response to reviewers of the paper Postmidnight equatorial plasma irregularities on June solstice during low solar activity – a case study

Anonymous Referee #1 Claudia M. N. Candido et al. claudia.candido@inpe.br

I would like to thank to the referres by the several usefull and respectfull comments and suggestions which contributed to the improvement of the paper.Please, find below the responses and changes highlighted in red here and in the manuscript.

[Figure]

Comment #1: Answer: Thank you. We agree that the reference by Y. Otsuka (2018) is relevant to the topic. We have included it in the revised paper.

Comment #2 Answer: We understand that the reported spread-F, as seen in the ionogramsis unusual with relation to the well-known sunset spread-F, since they appear at the higher frequency edge and progressively evolutes to a mixed (range and frequency) spread-F. Earlier reports showed that PMIs occur during very quiet geomagnetic conditions in June solstice and during low solar flux conditions as it is discussed in the present case. They are commonly reported as as FAIs in the observations taken from coherent and incoherent radar (Otsuka et al., 2009; Yokohama et al, 2011; Nishioka et al., 2012; Dao et al., 2017; Zhan et al. 2018), as well as are observed as depletions in the plasma density taken from satellite measurements (Dao et al., 2013; Yizengaw et al., 2013) or mild frequency spread-F seen in ionograms taken from off-equator stations. From our knowledge our PMIs observations were made using ionosondes and airglow simultaneously for the first in Brazil in the context of spread-F morphology and evolution.

Comment #3 Answer. Sure. This hypothesis is plausible and it was considered. However, there are no depletions in the OI 630.0 nm images neither spread-F in ionograms earlier in the night.

Comment #4 Answer. The anomalous patterns are related with its first appearance of spread-F echoes at the higher frequency edge of F-layer trace. Generally, the first spread-F echoes can appear as satellite trace at post sunset times or in the lower frequency edge. This anomalous pattern could be addressed as spur traces in the ionograms. Westward drifts are not rare, as mentioned by Otsuka et al., 2008. In Brazil westward depletions were observed during low solar activity and associated with previous depletions drifting eastward by Paulino et al., 2011. It is not the case studied in this work.

Comment #5: Answer. Please, see the answer 4.

Comment #6: Answer. It is currently accepted that a combination of factors can be responsible by the generation of PMIs or of these transient irregularities, especially owing the quiescent ionosphere. I personally agree that simulations could be useful to better investigate PMIs. However, we are performing analysis of other recent cases using the same instrumental approach and the simulation will be considered. Anonymous Referee #2

General comments: Presented in this manuscript is an analysis of an event in which postmidnight equatorial plasma irregularities occurred over Brazil during the June solstice of 2011. The peculiar feature of this event is the fact that the irregularities drift from east-to-west, as opposed to west-to-east. Such a phenomenon is typical of a geomagnetic storm, in which the isturbance dynamo results in a westward thermospheric wind at the equator that has ny equatorial plasma bubbles (and the plasma irregularities therein) imbedded among the neutrals. A good level of observations supports the overall conclusions in this work, including ionosondes, Fabry-Perot Interferometers and all-sky cameras.

The potential role of atmospheric gravity waves and background neutral winds are discussed as potential candidates for the presence of plasma irregularities during this particular evening. It is concluded that departures from the typical thermospheric wind system could be to blame for the increase in R-T growth conditions. The manuscript is mostly very well written and is easy to follow, and the examination and discussion of observations presented in the results section is relevant to the field and is acceptable for publication in Ann. Geophys. in the view of this referee, following the consideration of a few minor comments and suggestions, as detailed below.

Specific comments:

1. Line 37 – F10.7 values need units Thanks. It was included in line 35.

2. Lines 59-60 – Please be more specific than "distinct longitudinal sectors"; where do these PMIs occur? Thanks. It was highlighted in lines 121-124.

3. Line 123 – again, please specify which longitudinal sectors I agree. It was done in lines 121-124.

4. Lines 221-222 – "appeared during these oscillations" This feature isn't obvious in Fig 2. Can you please elaborate? Yes, sure. It was done in the lines 224-225.

5. Lines 238-239 – FZ and SL are indicated in all panels, are they not? Yes. Please, see lines 234-235 and Figure 02 and its caption.

6. Lines 239-240 – These two depletions are particularly difficult to see, particularly for readers unfamiliar/inexperienced in examining all-sky-camera data. Is there a way to make these features more obvious? Please, see the modified figure3 and caption. The dark regions passing over FZ and CZ are the signatures of plasma depletions. Also, it was mentioned between lines 241-245.

7. Figure 4 – Axes on these plots require labelling (i.e., N-E-S-W) Thanks. Please, see the changes in the figure 4.

8. Lines 260-261 – there is a rather sparse distribution between 2 and 5 LT as well, is there not? Related to this figure, are the two color panels on the left and right indicate the directions on the left and right of the plot, respectively? This isn't entirely clear.Thanks. Indeed, the time of the main spread-F echoes is around ∼23:00 LT to 01:00 LT, which are 02:00 and 04:00 UT, respectively.Please, see lines 263-265.The color code is clarified in lines 265-266.

9. Lines 265-266 – are the echoes in the NNE direction also at 23:30 LT and 415 km? The presence of these echoes is not clear to this reader. Yes. Thanks. We verified and clarified that the zonal distance is ∼ 320 km. See line 265-268.

10. Line 280 – the maximum upward velocity appears closer to 30 m/s to this reader. That is true. Thanks. It was fixed. Zonal drift <60 m/s and upward drift ∼30m/s.Please, see lines 285

11. Line 285 – the circles are not clearly visible, have they been removed? |V| are

represented by concentric circles with 25 m/s steps. See the modified text in line 290.

12. Lines 299-301 – more discussion/description is needed for the wave structures in Figure 7. These features are highlighted and discussed later in the discussion section, but they should first be highlighted here.

Many thanks. We modified the text to clarify this point. Please, see lines 305-322.

13. Lines 320-322 – The authors should specify here what makes this event "distinct"; i.e., they should mention their non-customary westward propagation. Please, see lines 343-346.

14. Figure 8 caption – "The shaded region is the monthly average" is rather misleading. I assume that the shaded region indicates the mean +/- one standard deviation. If this is the case, both the caption and the manuscript text (i.e., lines 306-307) should be clarified. Yes, thanks. It is corrected now in the figure 8 caption and in the text in lines 327-328.

Technical corrections:

1. Line 52 – "small-scale" and "large-scale" Thanks. It was done. Line 49

2. Line 157 – "small-speed" Thanks. It was done. Line 158.

3. Line 186 – "that" instead of "which" Thanks. It was done. Line 188.

4. Line 190 – "late-time" Thanks. It was done. Line 192.

5. Line 199 – "night and low" Thanks. It was done. Line 201.

6. Line 201 – "low-latitude" Thanks. It was done. Line 203.

7. Lines 279-280 – "|V| represents the zonal drift Doppler velocities are less than 50 m/s" does not read well. Please reword. Yes, thanks. It was corrected in line 285.

8. Lines 310-311 – Sentence beginning with "Additionally" also does not read well. Please reword. Thanks. It was done.

9. Line 321 – "low-latitude" Thanks. It was done. Line 342.

10. Line 330 – "post-sunset" Thanks. It was done. Line 354.

11. Lines 386-387 – "However, it is observed a secondary occurrence peak: : :" does not read well. Please reword. Thanks. It was done. Lines 411-413.

12. Line 390 – "discussed", not "discussing" Thanks. It was done. Line 415.

13. Lines 437-438 – "This condition leads to a negative RT-instability growth rate." Thanks. It was done. Line 459.

14. Lines 445-446 – It isn't clear to this reader what the authors mean by "may hands out". Please rephrase. It was a mistake. Thanks. It was fixed in line 467.

15. Line 462 – "Low-latitude" Thanks. It was done. Line 483 and 488.

16. Several references listed in the references section do not appear within the manuscript (e.g., Abdu et al., 1981b; Abdu et al., 1982; Bastia et al., 2004; and Carter et al., 2013, there could be others). The authors are encouraged to make sure that each paper listed has been cited at an appropriate location within the manuscript text. Also, Dao et al., 2016 appears in the text as "Dao et al., 2017".Dao et al 2016. See line 75. I agree. The list of references was carefully revised and fixed.